# New Approaches for the Treatment of AML beyond the 7+3 Regimen: Current Concepts and New Approaches

**DOI:** 10.3390/cancers16030677

**Published:** 2024-02-05

**Authors:** Jaime L. Roman Diaz, Mariola Vazquez Martinez, Farhad Khimani

**Affiliations:** Moffitt Cancer Center, Bone Marrow Transplant and Cellular Immunotherapy, Tampa, FL 33612, USAmariola.vazquezmartinez@moffitt.org (M.V.M.)

**Keywords:** AML—acute myelogenous leukemia, WHO—World Health Organization, ELN—European Leukemia Network, ICC—International Consensus Classification, NGS—next-generation sequencing, MDS—myelodysplastic syndrome, MRD—minimal residual disease, allo-HCT—allogeneic stem cell transplant, PBSCs—peripheral blood stem cells, CBU—cord blood unit, PTCy—post-transplant cyclophosphamide, GVHD—graft-versus-host disease, OS—overall survival, RFS—relapsed free survival, NRM—nonrelapse mortality, LDAC—low-dose cytarabine

## Abstract

**Simple Summary:**

Acute myelogenous leukemia is an aggressive heterogeneous hematologic malignancy, and its diagnosis, classification, and treatment have undergone significant changes in the last few years. As recently as 2022, the classification of the disease changed due to the emphasis on utilizing molecular technologies, which improved the understanding, prognosis, and treatments for AML. Ultimately, the possibility of a cure with these new treatment modalities is not possible for most patients. Allogeneic stem cell transplant is an alternative that can offer adequate disease control or a possible cure. Still, not all patients qualify for or benefit from transplant due to the disease’s risk classification or their age, performance, and comorbidities. We will describe all of this further in this article.

**Abstract:**

Fifty years have passed since the development of the first chemotherapy regimen for treating acute myelogenous leukemia (AML), with the approval in 1973 of the cytarabine daunorubicin (7+3) regimen. Until recently, patients diagnosed with AML had very limited treatment options and depended primarily on chemotherapy in combinations, doses, or schedules of the same drugs. Patients with advanced age, comorbidities, or relapsed or refractory disease were left with no effective options for treatment. New advances in the understanding of the biology and the molecular and genetic changes associated with leukemogenesis, as well as recent advances in drug development, have resulted in the introduction over the last few years of novel therapeutic agents and approaches to the treatment of AML as well as a new classification of the disease. In this article, we will discuss the new classification of AML; the mechanisms, actions, and indications of the new targeted therapies; the chemotherapy combinations; and the potential role of cellular therapies as new treatment options for this terrible disease.

## 1. Introduction

Acute myelogenous leukemia (AML) is a type of blood cancer characterized by the increased proliferation of myeloid blasts in the bone marrow, blood, and other organs or tissues [1]. Its prevalence is higher in adults, and it is the most common cause of mortality from leukemia in the United States. The average age at diagnosis is 68 years, with a substantial proportion of patients, over half of the cases, diagnosed over the age of 65 years [2,3]. The incidence of AML is increasing with the aging of the population. 

The classification and treatment of AML has undergone notable changes over the past few decades since the introduction of the first chemotherapy regimens like the cytarabine and anthracycline induction treatment known as the 7+3 regimen. This new understanding, moved by important advances in genetic and molecular science, has led to the emergence and revision of new classifications and the development of targeted treatment approaches that have revolutionized the field in the past few years. These new classifications enhance the accuracy of diagnosis and have significant implications for individuals with low-blast-count AML to have further opportunities in clinical trials and novel targeted agents early on in their disease progression. In this article, we want to summarize the changes to the classification and diagnosis of AML as well as the new treatment modalities for this disease, including stem cell transplantation. 

## 2. Classification and Diagnosis

As is well-known, the diagnosis and workup of AML involves the performance of a bone marrow aspirate and biopsy. This analysis includes flow cytometry, cytogenetics, and molecular studies in the form of next-generation sequencing (NGS). All this new in-depth molecular and genetic analysis helps find specific gene mutations that help in the classification of the disease and are needed in predicting prognosis and guiding therapeutic decisions for AML patients.

Going back to the 2016 WHO criteria, the definition of AML referred to the findings of 20% blasts in bone marrow or in the peripheral blood or disease outside the marrow. A diagnosis of AML was possible in cases with fewer than 20% blasts if it was accompanied by AML-defining genetic abnormalities, such as *PML::RARA*, *RUNX1::RUNX1T1*, and *CBFB::MYH11* [4].

The classification of AML has continued to evolve, with an emphasis on integrating clinical, morphologic, immunophenotypic, genetic, and molecular parameters in the classification to provide a more precise and evidence-based classification of AML. The integration of these parameters achieves personalized and targeted approaches in the diagnosis and management. It also helps design clinical trials that aim at the fundamental genetic and molecular drivers of AML. 

In the 2022 version of the WHO classification, the prior criteria of 20% blasts in peripheral blood or bone marrow is omitted for AML with defining genetic abnormalities. An exception to this is AML with *BCR::ABL1* fusion and AML with *CEBPA* mutation, which still require 20% blasts.

Essential changes in the new WHO classification also concern the AML–myelodysplasia-related subtype (previously called AML with myelodysplasia-related changes). Essential diagnostic criteria include a blast count of at least 20% in peripheral blood or bone marrow, and a mutation-based definition has been introduced [5].

The 2022 European Leukemia Network (ELN) established a risk classification by genetic analysis on diagnosis [6,7]. This model helps classify the patient’s risk of relapse and helps to coordinate the best treatment approaches. See Table 1.

Another important aspect is that the ELN highlights the significance of identifying minimal residual disease (MRD) by checking next-generation sequencing (NGS) or flow cytometry after initial therapy. Also, it recognizes hereditary (germline) predisposition to hematopoietic malignancies and offers guidelines for testing for these disorders. This helps to establish prognosis and subsequent treatment options for these patients, including the need for a future bone marrow transplant [7].

## 3. Treatment of AML

### 3.1. Young and Fit Patients

An intensive treatment approach is employed for young and physically fit patients, combining cytarabine and anthracycline chemotherapy like daunorubicin or idarubicin, called the “7+3” regimen. Daunorubicin is administered at 60–90 mg/m^2^ doses for three days and cytarabine at 100–200 mg/m^2^ for seven days [8]. This regimen carries the risk of cytopenia, with potentially life-threatening complications like infections and bleeding. Adverse effects include nausea, vomiting, mucositis, hair loss, and diarrhea. A flulike syndrome with rash and fever can be related to cytarabine; on the other hand, daunorubicin can lead to infusion reactions and decreased cardiac function. 

For acute myeloid leukemia (AML) with CD33-positive cells, gemtuzumab ozogamicin (GO) is considered beneficial, particularly in favorable or intermediate-risk AML cases [9]. GO is a monoclonal antibody against CD33 conjugated with calicheamicin. Randomized trials have resulted in mixed results. But a meta-analysis reported better outcomes when adding GO to a 7+3 regimen in favorable and intermediate-risk cytogenetics (6 years OS of 76 vs. 55% [OR 0.47, CI 95% 0.31–0.74] and 39 versus 34% [OR 0.84, CI 95% 0.75–0.95%], respectively) [9]. Other trials, like the multicenter phase 3 ALFA-0701 trial, reported superior event-free survival (EFS) with GO + 7+3 compared to 7+3 alone; however, no difference in OS was found [10,11]. In a study called AMLSG-09-09, patients diagnosed with *NPM1*-mutated AML were treated with idarubicin, cytarabine, etoposide, and all-trans-retinoic acid, with or without GO. The use of GO resulted in a lower cumulative incidence of relapse and a more significant proportion of undetectable measurable residual disease (MRD). However, more deaths were attributed to infections, and there was no noticeable difference in EFS [12,13]. However, GO routine use is curtailed due to the risk of sinusoidal obstruction syndrome, especially when most patients proceed to transplantation. Interestingly, the phase 3 ALFA-0701 trial showed that a fractionated dose of GO as part of the induction regimen for AML did not increase excess post-transplant VOD or mortality [14].

Midostaurin, a first-generation *FLT3* inhibitor, is approved for use with the 7+3 regimen during induction therapy. It is taken orally at 50 mg twice daily on days 8 through 21. The RATIFY trial showed that midostaurin improved overall survival (hazard ratio for death, 0.78; one-side *p* = 0.009) and EFS (hazard ratio for event or death, 0.78; one-side *p* = 0.002) in *FLT3*-mutated AML that received 7+3 and midostaurin as compared to those treated with 7+3 alone [15]. Quizartinib is another drug that has shown better survival when added to 7+3 induction therapy than adding a placebo to 7+3 in the phase 3 QuANTUM-First trial [16]. Patients in this trial received consolidation therapy with HiDAC + either quizartinib or placebo, allogenic HCT, or both, followed by continuation of single-agent quizartinib or placebo for up to 3 years. Median OS was 32 months versus 15 months, respectively. However, it was also shown that there was a higher rate of fatal events in the quizartinib arm [16]. It is important to mention that quizartinib has no inhibitory activity in FLT3-TKD-positive AML.

The question is whether patients with FLT3 mutation should receive post-HCT maintenance therapy. In a phase 3 multicenter Chinese trial, sorafenib maintenance after transplant for a patient with *FLT 3-ITD* mutations was associated with long-term survival (72.0% [95% CI 62.1–79.7] vs. 55.9% [45.7–64.9]; hazard ratio 0.55, 95% CI 0.34–0.88; *p* = 0.011) and with reduced relapse rates (15.0% [8.8–22.7] vs. 36.3% [27.0–45.6]; 0.33, 0.18–0.60; *p* = 0.0003) [17]. 

It is still not clear if maintenance therapy after HCT is recommended. Sorafenib has multiple toxicities, which could contribute to poor adherence to therapy. Another trial, the BMT CTN Protocol 1506, which is a phase 3 trial of gilteritinib as maintenance therapy after allogenic transplantation in patients with *FLT3-ITD* AML, seems to show some benefit with the use of gilteritinib. Results are still pending, but it will help us provide information about which patient may or may not benefit from maintenance therapy [18].

### 3.2. AML with Myelodysplasia-Related Features (AML-MRC) and Treatment-Related AML (t-AML) 

CPX-351 includes a liposomal combination of daunorubicin and cytarabine, expressly indicated for patients with t-AML and AML-MRC. The phase 3 trial played a pivotal role in CPX-351’s approval, showing a notable extension in survival for older adults with newly diagnosed secondary AML compared to the standard 7+3 regimen [19]. Median overall survival (OS) was superior in CPX-351-treated patients compared to those on 7+3 (37% versus 26% and 9.3 months versus 6 months). CPX-351 showed a prolonged time for recovery in neutrophil and platelet counts compared to the conventional 7+3 therapy (35 vs. 29 days and 36.5 days vs. 29 days, respectively). Early mortality appeared to be lower with CPX-351 in comparison to 7+3 therapy. Furthermore, there was a discernible trend showing that patients on CPX-351 were able to progress to allogeneic marrow transplant.

### 3.3. Older and Frail Patients

Frail and elderly patients with AML previously faced limited treatment options, primarily relying on supportive measures with a grim prognosis. Monotherapy involving hypomethylating agents (HMAs), like azacitidine/decitabine, had been employed with unclear benefits. A study by Kantarjian et al. showed no significant difference in the increase in median OS with decitabine (7.7 months; 95% CI, 6.2 to 9.2) vs. treatment choice (5.0 months; 95% CI, 4.3 to 6.3; *p* = 0.108) [20]. 

The landscape has significantly transformed with the introduction of new therapeutic combinations and the incorporation of venetoclax, heralding a revolutionary shift in treating AML. Venetoclax, a BCL-2 inhibitor, induces apoptosis in acute myeloid leukemia (AML) cells. Its approval, in combination with hypomethylating agents (HMAs) or low-dose cytarabine (LDAC), is specifically for older patients with AML. Results of the phase 3 VIALE-A trial showed that the combination of azacitidine/venetoclax improved responses and overall survival (OS) compared to azacitidine alone. For patients 75 years and older or with significant comorbidities, the combination showed a higher composite complete response (CR) rate of 66.4% versus 28.3% (*p* < 0.001). Additionally, the median OS was extended to 14.7 months from 9.7 months, with a hazard ratio (HR) of 0.66 (95% CI 0.52–0.85, *p* < 0.001) [21]. However, in the VIALE-C trial, venetoclax was added to LDAC versus LDAC monotherapy. The primary OS endpoint was not achieved due to diminished power and the inclusion of patients who received HMA as prior therapy in the VIALE C trial. But in the post hoc analysis with 6 months of added follow-up, OS advantage was seen (8.4 vs. 4.1 months, HR 0.7, 95% CI 0.5–0.98, *p* = 0.04). These treatment regimens are currently recognized as the established standard for older individuals or those who are not considered suitable for more intensive chemotherapy.

IDH1 inhibitors have gained approval as a standalone treatment for relapsed or refractory (R/R) IDH1-mutated AML and newly diagnosed AML poor candidates for intensive chemotherapy. Based on the phase III AGILE trial, the FDA has approved the combination of ivosidenib and azacitidine specifically for newly diagnosed *IDH1*-mutated patients aged 75 and above or those with comorbidities.

In this trial, patients received azacitidine plus ivosidenib or azacitidine alone. The study showed a better event-free survival (EFS) in patients treated with azacitidine and ivosidenib compared to those receiving azacitidine alone (hazard ratio [HR] 0.16, 95% confidence interval [CI] 0.16–0.69). Median overall survival (OS) was also higher in patients who received ivosidenib compared to azacitidine alone (24 vs. 7.9 months). Febrile neutropenia was higher in the azacitidine-plus-placebo group compared to the ivosidenib-plus-azacitidine group (34% vs. 28%, respectively). Differentiation syndrome occurred in 14% of patients receiving ivosidenib and azacitidine and 8% of patients receiving azacitidine plus placebo [22].

IDH 2 inhibitors like enasidenib have also been used in clinical trials in newly diagnosed AML. In a phase 2/1b trial, the efficacy of enasidenib as monotherapy was seen in patients with the mutation and an age equal to or above 60 years. Patients who received enasidenib monotherapy had a composite complete response of 46%. Additionally, if patients did not have a complete remission or had a CR with incomplete blood count recovery, they could enroll in the phase 1b cohort where azacitidine was added. Of the patients who transitioned to the phase 1b trial, the cCR rate was 41%. This trial showed a risk-adapted approach, and it also showed the efficacy of enasidenib monotherapy [23].

Glasdegib, classified as a hedgehog pathway inhibitor, targets an overexpressed pathway in myeloid leukemia cells. Building on this discovery, a clinical evaluation was conducted to evaluate glasdegib in combination with low-dose cytarabine (LDAC) compared to LDAC alone. The study focused on patients over 75 years old who were newly diagnosed with AML and were not suitable for intensive treatment because of age or significant comorbidities.

In this phase II trial, patients were randomized to receive 100 mg of oral glasdegib daily for 28-day cycles. LDAC was given at 20 mg twice daily for 10 days within a 28-day cycle. The primary endpoint showed a median OS of 8.8 months with glasdegib plus LDAC and 4.9 months with LDAC alone (hazard ratio, 0.51; 80% confidence interval, 0.39–0.67, *p* = 0.0004). Notably, the complete remission rate was significantly higher in the glasdegib-plus-LDAC arm at 17% compared to 1% in the LDAC-alone arm (*p* < 0.05) [24]. The most common side effects seen in the glasdegib-plus-LDAC cohort were pneumonia and fatigue. The study provides information on the benefits of combining glasdegib with LDAC in patients who are not good candidates for more intensive chemotherapy approaches.

### 3.4. Postremission Therapy

Once remission is obtained, consolidation therapy is essential for an excellent long-term outcome and a possible cure. The choice of treatment is guided using the European Leukemia Network (ELN) risk score as a guide. So, patients with a favorable ELN risk score undergo three to four cycles of high-dose chemotherapy, specifically, intermediate- to high-dose cytarabine consolidation. However, patients with intermediate- or adverse-risk scores are referred for allogeneic stem cell transplantation [25]. 

These recommendations are based on the high risk of relapse and mortality associated with these higher-risk cases. Patients with intermediate-risk or adverse-risk AML who cannot proceed to transplant have options like oral azacitidine or CC-486. The phase 3 Quazar AML-001 trial showed longer OS with CC-486 than with placebo (24.7 months vs. 14.8 months; *p* < 0.001) and longer relapse-free survival (10.2 months vs. 4.8 months; *p* < 0.001) [26]. 

### 3.5. Relapsed/Refractory Disease

The five-year OS for relapsed or refractory (R/R) disease is approximately 10% [27]. The primary aim for individuals experiencing R/R disease is to reach a second complete response. Depending on the specific case, various strategies can be employed. If a patient has previously undergone an allogeneic transplant in complete remission (CR1), options may include immunosuppression tapering to enhance the graft-versus-leukemia effect, infusion of donor T cells (DLI), immunotherapy, or a second allogeneic transplant if possible. For those who did not undergo an allogeneic transplant in CR1, considering it to achieve a second remission becomes a practical choice. Additionally, targeted therapies have gained approval for relapsed/refractory AML.

IDH mutations happen in 15–20% of AML cases. The *IDH1* inhibitor ivosidenib, as mentioned before, has gained approval for patients with relapsed or refractory *IDH*-mutated AML. In the phase 1 trial, a daily dose of 500 mg of ivosidenib demonstrated durable remission and molecular remission in some patients, including complete remission [28].

Later phase 1b-II trials of ivosidenib with venetoclax, plus or minus azacitidine, in IDH1-mutated myelodysplastic syndrome (MDS), newly diagnosed AML, or relapsed/refractory AML, yielded impressive outcomes. Sixty-three percent of AML patients achieved minimal residual disease (MRD) negativity. The 24-month overall survival (OS) rates were 50% in relapsed/refractory AML and 67% in newly diagnosed AML. While these results are promising, further comparisons to the standard of care or investigations in larger populations are necessary to validate the trial’s potential [29].

Olutasidenib has recently gained approval for relapsed or refractory (R/R) *IDH1*-mutated acute myeloid leukemia (AML). A phase I–II trial was an open-label, single-arm, multicenter clinical trial of relapsed or refractory *IDH1*-mutated AML. Olutasidenib was administered at a dosage of 150 mg PO twice daily until disease progression, toxicity, or introduction of an allogeneic transplant. The results showed a complete remission (CR) rate of 32%. The median overall survival (OS) for patients with R/R AML was 8.7 months with monotherapy and 12.1 months with combination therapy. Among the 147 treated subjects, 34% of transfusion-dependent patients became independent of red blood cell (RBC) and platelet transfusions [30]. Olutasidenib has a black box warning related to differentiation syndrome. This shows the importance of close monitoring and management of these patients. 

On the other hand, enasidenib is an *IDH2* inhibitor approved for patients with relapsed and refractory AML with *IDH2* mutation. Its approval came after the results of a phase 2 single-arm study. The results showed an ORR of 40.3% with a median duration of 5.8 months and a median OS of 9.3 months. For patients with complete remission, the OS was 19.7 months [31]. Enasidenib has been compared with lower-intensity therapies, including LDAC, HMA, or best supportive care, in a post hoc IDENTIFY phase III trial analysis, showing better CR rates and OS [32].

FLT3 inhibitors are another class of drugs that have a key role in targeted therapy for relapsed and refractory acute myeloid leukemia (AML). Gilteritinib was approved for patients with relapsed/refractory *FLT3*-mutated AML based on the ADMIRAL trial. In this trial, the OS in the gilteritinib arm was 9.3 months, showing a marked improvement compared to the standard chemotherapy group, with a median OS of 5.6 months (hazard ratio for death, 0.64; 95% confidence interval [CI], 0.49 to 0.83; *p* < 0.001). It also showed fewer side effects compared to the chemotherapy group [33]. Grade 3 or higher side effects in the gilteritinib arm were neutropenic fever (45.9%), anemia (40.7%), and thrombocytopenia (22.8%) [33]. This shows how effective gilteritinib is as a targeted therapy against *FLT3*-mutated AML, offering better survival with a more tolerable side effect profile. 

Other trials have investigated the combination of gilteritinib with venetoclax in patients suffering from relapsed/refractory (R/R) AML. For example, one trial resulted in a modified complete response (mCR) rate of 75%, with 36% of the subjects achieving a morphological leukemia-free state [34].

Combinations with hypomethylating agents (HMA), venetoclax, and gilteritinib in the *FLT3* relapsed setting were explored. In one trial, the triple therapy showed high response rates, with an overall response rate (ORR) of 67% and a median overall survival (OS) of 10.5 months [35]. On the other hand, combining these three agents comes with a risk of significant hematological toxicity, with a dose reduction of gilteritinib needed. This regimen can potentially be used as a bridge prior to transplant in certain patients. 

Other triplet combinations included HMA + venetoclax and *IDH2* inhibitor enasidenib in a study by Sangeetha et al. It was seen that patients with R/R AML who were not candidates for intensive chemotherapy at a median follow-up of 11.2 months had a median OS that was not reached and a 6-month OS of 70% [36]. Although the number of patients was small, the therapy did improve outcomes.

The combination of venetoclax and FLAG-IDA in relapsed/refractory acute myeloid leukemia (R/R AML) achieved complete remission (CR) rates from 61% to 75% and a one-year overall survival (OS) of 68% [37]. This combination is highly myelosuppressive. During the trial, adjustments were made to the venetoclax administration period, reducing it from 14 days to 7 days. Real-world data on these combinations revealed that approximately half of the patients experienced invasive infections, emphasizing the need for vigilant monitoring and swift intervention [38]. It is important to note that allogeneic transplantation is an essential part of this treatment approach for ensuring long-term survival in this population of intermediate- to high-risk patients with relapsed refractory disease. Please see Table 2 for summary classification of AML therapies.

### 3.6. Investigational Therapies

AML remains a diagnosis marked by uncertain outcomes, occasionally yielding unsatisfactory results. Numerous trials have explored targeted therapies and immunotherapies as potential treatments for AML. Some of these trials have exhibited promising results, extending hope even for patients with *TP53*-mutated AML or AML with MLL rearrangements.

Patients with MLL rearrangements have been associated with having an adverse prognosis. Recently, the scaffold protein menin has been a subject of study. Menin plays a crucial role as it binds to *KMT2A*, and this interaction is essential for *KMT2A*’s function. Currently, there are drugs designed to interfere with the binding of menin to *KMT2A*. SNDX 5613 has shown promising results (Pivotal Augment-101 trial), achieving a 63% overall response rate and median OS at the time of data cutoff of 8.0 months [39]. Another drug, KO539, has also demonstrated some efficacy in this regard [40]. In a phase I/II study by Ghayas et al., although it only had eight patients, showed 71% CR rates. The study added menin inhibitor revumenib in patients harboring *KMT2A* or *NUP98* rearrangements or *NPM1* mutations. The results, although very early and in a very small population, indicate a good efficacy of this combination in patients with R/R AML [41].

Magrolimab, an anti-CD47 monoclonal antibody, has displayed antileukemia activity. A study focusing on patients with *TP53* mutations investigated the outcomes of combining magrolimab with azacitidine. This combination was assessed in a phase 1b trial involving newly diagnosed AML patients who are not good candidates for aggressive chemotherapy. The results revealed a 71% response rate in patients with TP53 mutations, with 45% achieving complete remission (CR). The median estimated overall survival (OS) was reported to be 10.8 months [42]. Unfortunately, the phase III trial ENHANCE 2 and ENHANCE 3 trials were discontinued due to futility based on planned analysis.

Other therapies have shown promise, such as sabatolimab (MBG453), a humanized antibody targeting TIM3—an inhibitory checkpoint in leukemia but not in normal stem cells. In a phase Ib trial involving patients with AML and high-risk MDS, sabatolimab demonstrated a 41.2% response rate in individuals with newly diagnosed AML [43]. A phase II trial evaluated sabatolimab in combination with azacitidine and venetoclax. This trial involved two dose levels (400 mg and 800 mg) of sabatolimab, and the safety profile observed is comparable to the reported safety profile of venetoclax and azacitidine [44].

Uproleselan (GM-1271), an E-selectin inhibitor, has shown efficacy in patients with R/R AML treated with the MEC regimen (mitoxantrone, etoposide, and intermediate-dose Ara C) alongside uproleselan. In a phase I/II trial, a complete response (CR) was achieved in 35% of cases, and the median OS was reported as 8.8 months [45]. Currently, a phase III trial (NCT03616470) is actively assessing the impact of including uproleselan in chemotherapy in patients with relapsed/refractory AML.

AML treatment is an evolving field, with new therapies being explored. Bispecific antibodies and chimeric antigen receptor (CAR) T-cell therapies with potential targets such as CD123, CD33, and CD70 are under study. These innovative approaches hold promise and represent ongoing efforts to find effective treatments for AML. Indeed, the challenge with targeting antigens like CD33 and CD123 in AML therapy lies in their abundant presence in normal stem cells and progenitor cells, risking profound myelotoxicity. Additionally, AML is a very heterogeneous disease and varies between patients depending on chromosomal abnormalities and mutations. It also creates a very immunosuppressive tumor microenvironment containing various cell types like regulatory T cells, macrophages, dendritic cells, and myeloid-derived suppressor cells that suppress T-cell activity and decrease CAR T-cell responses [46]. Addressing this concern, a current approach under investigation involves the use of CRISPR/Cas9 technology to edit out the CAR target antigen from a donor allograft. This innovative strategy minimizes the impact on normal hematopoietic cells while effectively targeting AML cells [47].

### 3.7. Allogeneic Bone Marrow Transplant 

Allogeneic stem cell transplant (allo-HCT) offers the only potential cure for some AML patients. The long-term survival rate of a patient who undergoes allo-HCT can vary between 10% and 70%, considering factors like patient demographics, clinical variables, genetic abnormalities, and measurable residual disease [48]. Many factors influence the decision for a patient to undergo a bone marrow transplant. If a patient has “high-risk” or “intermediate-risk” cytogenetics or has the presence of MRD after induction, then a transplant in the first complete remission (CR1) or achievement of maximum response is recommended. However, if the cytogenetic risk is “favorable,” a transplant in CR1 is usually not advised [49,50,51]. Several studies support the benefits of allogeneic stem cell transplant in adverse- and intermediate-risk cases but not in favorable-risk cytogenetics.

A meta-analysis of 24 trials involving over 3500 transplant recipients found that patients with adverse- or intermediate-risk cytogenetics had significantly better outcomes after allo-HCT, particularly regarding relapse-free survival (RFS), than those with favorable-risk cytogenetics. The hazard ratio (HR) for RFS was 0.69 (95% CI: 0.57 to 0.84) for adverse risk and 0.76 (95% CI: 0.68 to 0.85) for intermediate risk. However, there was no significant benefit in RFS for favorable-risk cytogenetics, with an HR of 1.06 (95% CI: 0.80 to 1.42). Overall survival (OS) followed a similar trend as RFS based on cytogenetic risk [25]. 

Another essential factor to consider in the decision to proceed with an allogeneic stem cell transplant is persistent MRD (MRD+). This critical factor affects the prognosis of patients, regardless of their cytogenetic risk group [52,53,54]. In the AML17 clinical trial, for the *NPM1*-wt standard-risk subgroup, MRD+ after two cycles was significantly associated with poorer outcomes (OS, 33% vs. 63% MRD−, *p* = 0.003; relapse incidence, 89% when MRD+ ≥ 0.1%); transplant benefit was more apparent in patients with MRD+ (HR, 0.72; 95% CI, 0.31 to 1.69) than those with MRD− (HR, 1.68 [95% CI, 0.75 to 3.85]; *p* = 0.16 for interaction) [55]. 

Therapy-related acute myeloid leukemia (t-AML) and secondary AML (s-AML) have a significantly worse outlook compared to de novo AML and are classified as high-risk. Guideline recommendations are for allo-HCT in CR1 for eligible patients with t-AML or s-AML. Allogeneic stem cell transplant is recommended for AML patients based on their risk profile, with patients in the higher- to intermediate-risk classifications having the best benefits [54].

Although allo-HCT carries, in general, a lower risk of relapse and possible cure of leukemia, it has higher nonrelapse mortality (NRM) compared to chemotherapy alone. In surviving patients, graft-versus-host disease (GVHD) can lead to significant morbidity [56]. For this reason, it is important to consider patient- and disease-related factors, psychosocial circumstances, the patient’s age, and other comorbidities that are crucial in improving the outcomes of the transplant.

The choice of the conditioning regimen for transplant is important, as it has a significant impact on allo-HCT outcomes. The two main conditioning treatment modalities are myeloablative conditioning (MAC) and reduced intensity conditioning (RIC). Toleration of these regimens depends on factors like age, performance, and comorbidities. The randomized phase III BMT CTN 0901 study evaluated patients aged 18 to 65 years with <5% marrow myeloblasts to receive MAC (n = 135) or RIC (n = 137), followed by HCT from an HLA-matched donor. Results showed a significantly lower RI with MAC compared to RIC (15.9% versus 51%), resulting in higher 18-month OS (76.4% versus 63.4%) [57]. Long-term follow-up of BMT CTN 0901 showed consistent OS benefit with MAC (4-year OS: 65% versus 49%, *p* = 0.02) compared to RIC [58]. Per ASTCT consensus recommendations, in individuals under 65 years of age with AML in CR1, a myeloablative conditioning regimen (MAC) is preferred. On the other hand, in the case of older or less fit patients, reduced intensity conditioning (RIC) is an acceptable option [54].

Regarding donor selection, it is important to note that a fully HLA-matched related donor (MRD) is only available for approximately 25% to 30% of patients evaluated for allo-HCT. The probability of finding an 8/8 HLA-matched unrelated donor (UD) through the National Marrow Donor Program varies, with around 75% for whites of European descent and 16% for blacks and Hispanics [59]. If a fully matched donor is not available, good outcomes have been reported to alternative donor transplantation with haploidentical (haplo-HCT) and mismatched unrelated transplant (MMUD), which increases the availability of donors to more transplant-eligible patients and patients of various ethnicities since the chances of finding a matched related donor (MRD) or a matched unrelated donor (MUD) may be lower in these populations. For example, haplo-HCT with pTCY has shown comparable clinical outcomes to the matched unrelated donor (MUD) allo-HCT and has gained significant use in these cases. In addition, another benefit of haplo-HCT is that it has a lower risk of chronic graft-versus-host disease (cGVHD) with the addition of post-transplant cyclophosphamide (pTCY) [60,61,62,63].

## 4. Conclusions

It has been an exciting decade with many changes in the field of AML that have increased the arsenal of therapies and improved outcomes. We have seen the evolution of the treatment of AML from a standard chemotherapy induction with subsequent molecular classification, guidance of therapy based on the molecular makeup of the disease to targeted therapies, and potential new therapies with immune effector cells that harness the patient’s immune system to fight leukemia. Further ongoing research and trials will add more and lead to better outcomes in AML.

## Figures and Tables

**Table 1 cancers-16-00677-t001:** ELN risk classification at diagnosis.

Risk		
Favorable	Intermediate	High
**t(8;21)(q22;q22.1)/*RUNX1::RUNX1T1*** **inv(16)(p13.1q22) or t(16;16)(p13.1;q22)/*CBFB::MYH11*** **Mutated *NPM1* without *FLT3*-ITD** **bZIP in-frame mutated *CEBPA***	Mutated *NPM1* with *FLT3*-ITD Wild-type *NPM1* with *FLT3*-ITD t(9;11)(p21.3;q23.3)/*MLLT3::KMT2A* Cytogenetic and/or molecular abnormalities not classified as favorable or adverse	t(6;9)(p23;q34.1)/*DEK::NUP214* t(v;11q23.3)/*KMT2A*-rearranged (except partial Tandem duplication) t(9;22)(q34.1;q11.2)/*BCR::ABL1* t(8;16)(p11;p13)/*KAT6A::CREBBP* inv(3)(q21.3q26.2) or t(3;3)(q21.3;q26.2)/*GATA2*, *MECOM(EVI1)* t(3q26.2;v)/*MECOM(EVI1)*-rearranged −5 or del(5q); −7; −17/abn(17p) Complex karyotype > 3 abnormalities, monosomal karyotype 2, or more monosomies Mutated *ASXL1, BCOR, EZH2, RUNX1, SF3B1, SRSF2, STAG2, U2AF1, or ZRSR2*(not if occur in fav risk AML) Mutated *TP53*

Source: Adapted from [6].

**Table 2 cancers-16-00677-t002:** Current targeted therapies.

Target	Therapy
*FLT3* inhibitors	Frontline: midostaurine + chemotherapy. Quizartinib + chemotherapy. Relapse/refractory: gilteritinib. Maintenance following consolidation: quizartinib.
*IDH1* inhibitors	Ivosidenib: frontline in adults > 75 yo and/or with comorbidities in the R/R setting. Olutasidenib: in the R/R setting.
*IDH2* inhibitor	Enasidenib: R/R setting.
Anti-CD33 monoclonal antibody	Gemtuzumab ozogamicin: during induction therapy for CD33-positive AML with chemotherapy or as monotherapy in the R/R setting.
Secondary AML or t-AML	CPX-351: as induction chemotherapy for newly diagnosed secondary AML or t-AML.
BCL-2 inhibitor	Venetoclax: newly diagnosed AML in patients > 75 years old or with comorbidities in combination with HMA or LODAC.
Hedgehog pathway inhibitor	Glasdegib: for adults age > 75 who have comorbidities.

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
