# Peer review of "New Approaches for the Treatment of AML beyond the 7+3 Regimen: Current Concepts and New Approaches"

_cancers, 2024, doi:10.3390/cancers16030677_

Round 1
Reviewer 1 Report
Comments and Suggestions for Authors
Jaime Diaz and colleagues present a comprehensive review article entitled
“New approaches for the treatment of AML beyond 7+3”
considering important aspects of recently updated AML classification, aspects of approved induction regimens and reporting expected advances in targeted therapies for AML patients.
According to the title of this article, the manuscript should clearly focus on treatment aspects of AML and especially on new approaches. Otherwise, the title might be adapted, e.g. “Current concepts and new approaches …”.
Despite important aspects of at least three classification systems being clinically relevant for both prognostic stratification and the decision for a distinct induction regimen, I do not recommend to include such large tables just copying all these comprehensive ICC, WHO and ELN classifications.
In the FLT3 section, the authors describe the results of the RATIFY study and the QuANTUM-First trial without distinguishing between ITD and TKD mutations. It might be included that quizartinib has no relevant inhibitory activity in FLT3-TKD-positive AML.
The term “FLT3 inhibitors can be used for induction therapy in patients with FLT3 mutated AML …” is somewhat misleading because is should be seen as the SOC for this subset of AML patients.
Furthermore, the authors describe results from the Morpho study evaluating gilteritinib after alloSCT. The citation [15] should be checked because number 15 describes a publication with GO. With respect to TKI for FLT3-ITD AML patients undergoing alloSCT, the SORMAIN study should be considered in such a review.
In consideration of the title and presumably the aim of the authors to report on new approaches in AML therapy, the section describing investigational therapies is very short without expected details. For instance, menin inhibition is a very important and highly promising approach not only in KMT2Ar AML that earns much more attention in such a review. Recent data presented at ASH 2023 are available and part of is worth to be included here.
I suggest to create a figure summarizing current and future concepts of targeted therapies.
In addition, present figure 1 should be revised. In my view, the arrangement and rather simple design is more confusing than helpful to get a good overview of SOC and implementation of new concepts. Furthermore, this figure does not reflect clinical standards at some point (e.g. alloSCT for “ELN unfavorable” is missing, maintenance treatment without alloSCT using midostaurin or onureg is not included, HMA/VEN is not approved for patients eligible for induction therapy – at least in the EU).
Author Response
Thank you for taking the time to review this article. The modifications as per the recommendations are marked in yellow.
1- Title was modified based on your recommendation.
2-Tables were eliminated and a more direct explanation of the findings was given.
3- Recommendations regarding FLT3 inhibitors were followed.
4-Citation number 15 was revised.
5-Recent data on menin inhibitors from ASH 2023 was added
6-Figure on current therapies was added. Original Figure was removed.
Reviewer 2 Report
Comments and Suggestions for Authors
The review by Roman Diaz, et al, describes the current state of treatment for AML. This is a very large topic and the authors keep their review at a fairly basic level, discussing the most commonly used new therapies but not going very far into newer agents or chemotherapy regimens. Therefore the review is appropriate for a reader who is relatively unfamiliar with AML therapy and wants a general introduction. Specific comments to be addressed follow.
Line 37 – should it be 2016 not 2017?
Line 54 – the WHO 2022 criteria as presented in reference 6 actually do not lower the blast threshold to 10% for most genetic subtypes. Rather, they eliminate a minimum threshold, as described on page 1708 of that reference and as acknowledged in the title of Table 2.
Line 94 – adverse cardiac effects from anthracyclines include decreased left ventricular ejection fraction, even heart failure in severe cases.
Line 125 – reference 18 cited here may be an error as the reference is to a trial of CPX-351, not quizartinib.
Line 136 – reference 15 cited here also is incorrect. Please carefully check all citations.
Lines 174-187 – please also include a brief discussion of IDH2 mutated AML and use of enasidenib for newly diagnosed patients. DOI: 10.1182/bloodadvances.2023010563
Line 265 – reference 27 cited here also is incorrect. Please carefully check all citations.
Lines 272 – 277 – please include a discussion of other triplet combinations, meaning HMA + venetoclax + targeted inhibitor. Besides gilteritinib, triplets with IDH1/2 inhibitors are used routinely, and others are under investigation.
Lines 327-336 – In the discussion of immunotherapies like bispecific antibodies and CAR-T cells, please consider including a brief discussion of how these approaches are much more challenging in AML compared to lymphoid malignancies where they have been very successful. Besides the fact that there is a lot of overlap in antigens between AML and normal cells, as you have mentioned, the immune system in AML is often quite suppressed. This means that therapies that require functioning T cells, such as bispecific antibodies and CAR-T cells, are unlikely to be successful unless the underlying T cell dysfunction is also addressed.
Lines 348-350 – This sentence is confusing. It sounds like you are saying that patients with high risk or intermediate risk AML had better outcomes than those with low risk AML. Please clarify that high risk and intermediate risk patients who had a stem cell transplant did better than those who did not.
Line 360 – I doubt that the relapse incidence cited here was within 3 years of SCT. Based on the figures in reference 52, it looks like it is within 3 years from study entry. Please confirm and revise as appropriate.
References 58-61 are not found in the text. Please cite where appropriate or remove them from the references list.
Comments on the Quality of English LanguageThere are incomplete sentences and errors in punctuation, grammar and spelling. In most cases the intended meaning of the authors can still be understood.
Author Response
Thank you for reviewing the article. The recommended changes are marked in yellow. The lines have moved from the original manuscript.
1- WHO date changed to 2016.
2-Classifications and tables were modified and some removed as recommendation from the other reviewer.
3-Cardiad effect of anthracyclines was modified to add HF.
4-Reference 18, reference 15 and 27 were corrected. All references were revised.
5-Discusion on IDH2 inhibitor was added.
6-Discusion on triplet combinations was added.
7- Discussion on bispecifics was added.
8- Line 348 was clarified.
9-Line 360 was clarified.
9- Old references 58-61 were corrected.
Round 2
Reviewer 1 Report
Comments and Suggestions for Authors
The authors substantially improved this review.